# Turned Surface Monitoring Using a Confocal Sensor and the Tool Wear Process Optimization

Jozef Jurko [1], Martin Miškiv-Pavlík [1], Jozef Husár [1,*] and Peter Michalik [2]

1 Department of Industrial Engineering and Informatics, Faculty of Manufacturing Technologies, Technical University of Košice, Bayerova 1, 08001 Prešov, Slovakia

2 Department of Automotive and Manufacturing Technologies, Faculty of Manufacturing Technologies, Technical University of Košice, Štúrova 31, 08001 Prešov, Slovakia

* Correspondence: jozef.husar@tuke.sk; Tel.: +421-55-602-6415

**Abstract:** Laser scanning technology has been used for several years. Nevertheless, no comprehensive study has been conducted to prove that the application of confocal chromatic sensor (CCHS) laser technology is effective and suitable to verify the integrity parameters of machined surfaces in terms of cutting tool damage. In this paper, the optimization and effects of five factors (cutting speed, feed, depth of cut, attachment length of the workpiece, and tip radius) on the roundness deviation measured by CCHS and, at the same time, on the amount of wear on the back side of the cutting part of the tool were studied according to ISO 3685, which was measured with a microscope. The results obtained were evaluated using the gray relational analysis method (GRA), in conjunction with the Taguchi method, and the significance of the factors was demonstrated using the analysis of variance (ANOVA) method.

**Keywords:** confocal chromatic sensor (CCHS); cutting tool wear; turning; gray relational analysis; Taguchi method; ANOVA





## 1. Introduction

Machining (turning) still holds importance, and its replacement with other technologies has not always been successful. The ideal turning process can be characterized as the process of manufacturing a product following attachment. The ideal turning process is also a process through which we can manufacture a product in compliance with the specified criteria. The cutting tool plays a key role in understanding the behavior of machining operations [1]. We can also apply the ideal measurement during inspection. The ideal measurement of the product is a process through which we can measure the parameter following attachment and simultaneously measure the parameter during the attachment after the last operation of the cutting process in the working zone of the machine. The use of intelligent systems for identifying tool wear during turning was investigated by [2]. In this article, we present the results for the hypothesis that it is possible to apply CCHS to measure the quality parameters for a product after turning. The mentioned hypothesis is supported by previously conducted research with a triangulation laser sensor, and some results are presented in the article. The main thesis of this research is that we can use a laser beam to check the quality parameters after the cutting process. It is important to pay attention to the correct definition of the conditions for measurement with CCHS. In addition, this research focuses on the implementation of a new sensor for measuring the parameters of a machined surface.

When measuring the parameters of the machined surface, we identify the various errors that must be analyzed and evaluated realistically and these should be suppressed in the results as much as possible. In order to eliminate some measurement errors and also minimize the influence of the human factor, we verified a new-generation CCHS application. The laser sensor CCHS (CL-P070) from Keyence (Osaka, Japan) was used

for this research. Since the research results are important not only for theory but also for practice, measurement systems used in companies were used to measure the functional parameters. The main focus of the research was the use of the CCHS to measure the roundness deviation ($D_R$) and the analysis of the influence of the change in the roundness deviation on the flank wear parameter ($VB_N$) of the cutting insert. Another focus of the research was the design and optimization of the input factors during turning and the determination of their significance of influence, the verification of the holder design for the CCHS and the data transfer algorithm design for the CCHS. The obtained data will form the basis for the design of models for the implementation of laser sensors in the working zone of the machines. Another part of the investigation was the comparison of the measured values of roundness deviation with the measured values from ($D_R$) on the ROUNDTEST RA 120 device from Mitutoyo (Kanagawa, Japan). At the end of the investigation, a comparative test was carried out to verify the findings of the investigation and the effect on the optimization of the input factors in the interpretation of the importance of the proposed input factors and the comparison of data between the conventional and the progressive measurement system [3].

In the context of elimination, due to the human-caused errors in product control, new control methods are gaining attention in which a person merely performs the function of an operator. Several researchers have conducted research in this area. Zhou et al. used laser sensors for active control during the turning process and for the identification of defects [4]. Muszynski et al. studied laser sensors primarily for different types of control, be it operational or output control [5]. Similar studies were conducted by You et al. using blue laser technology, which has high measurement accuracy [6]. Laser sensors for position measurement are also widely used in the transportation industry. Yu et al. in their study used a confocal sensor to measure the thickness of transparent materials [7]. Another important application of laser sensors is the control of elements in the working area of the machine and the detection of geometry errors [8]. The advantages and disadvantages of laser sensors are presented in the study of Jaworski et al. [9]. Other authors have dealt with the conditions for using CCHS to measure machined surfaces. The main advantages of non-contact measurement systems based on optical methods are high scanning speed and measurement accuracy [10]. The proper setting of conditions for the use of CCHS was studied by Chen and Bai et al. [11,12]. The determination of the input factors is very important when using CCHS. In their research, Li, Berkovic, Yu et al. dealt with the analysis of the obtained data, the determination of the start of the measurement of the sensors, and the effect of the illumination angle on the surface [13–16]. Despite the various published studies on the application of laser sensors, there are few areas that have been explored. Machining technology and, in particular, the control of the accuracy of products through the use of laser sensors are among the areas that require experimentation to clearly define the conditions for their application. The authors presented the research results of several studies in which they verified and proposed the possibilities of using CCHS in machining. One of the most important factors in machining is the vibration in the working area of the machine and in the cutting zone. If the machine tool works within the prescribed vibration tolerances (established by the manufacturer or through diagnostic verification), we can effectively use laser sensors. The use of laser sensors to measure the roughness parameters of the machined surface was studied by Fu, Grochalski, Yuan, Liu and Wang et al. [17–21]. Non-contact measurement of surface roughness parameters was studied by Fu et al. [22].

When processing data obtained from measurements with sensors, it is important to analyze and possibly separate the data from errors due to various other influences. This factor is always important, especially when the measurements are taken on a machine that is not calibrated. In their study, Gao et al. present several algorithms to separate these errors [23]. Hrehova et al. describe the application of non-contact sensors using a neural network to detect points on the surface of the workpiece [24]. A new device to analyze the real-time deformation behavior of materials was studied by Singh et al. [25].

The study of the relationship between the parameters of surface integrity and tool damage is an area in which we cannot give an exact result. In another study, the authors were concerned with the study of tool damage [26,27]. The results of the indirect measurement of tool wear using the $VB_N$ parameter regarding the deviations from the average values were presented in the study of Jurko et al. [3]. The quality parameters of a machined surface often depend on the cutting tool damage, and it is influenced by factors and phenomena in the cutting zone, such as cutting conditions, cutting forces, vibrations, temperature, process medium, etc. Predicting the wear of the cutting part of the tool is particularly important, as pointed out by several researchers and according to Brillinger, Tabaszewski and Uhlmann, the methods and tools used for processing the obtained data are also important [28–37]. Wilkowski et al. developed a model to evaluate tool life [38]. Other authors have dealt with the effects of machined surface parameters on tool life [39,40].

From the study of the literature, it appears that the research on surface integrity with the application of new technologies generates new solutions that need to be studied. The study of the control of the roundness deviation parameter according to (ISO-12181-1), using CCHS on a reference sample of C45 steel, is the basis for other materials. A systematic study of the effects of process factors on machined surfaces proves that we can also influence the results of the control of the input factors to achieve the desired values by changing these factors. The way to improve surface integrity parameters is to optimize the combination of important input factors, such as cutting speed, feed, depth of cut, workpiece attachment length, and tip radius of the cutting edge. On the one hand, the input factors—the conditions of the machining process—and on the other hand, the CCHS factors must be matched. These input factors can be adjusted during machining, which is advantageous for automation and intelligent machining.

We can understand the GRA method as a system into which we input information that we know, and receive information that we do not know [41,42]. By combining the GRA method with the Taguchi method, we can achieve multi-optimization. The Taguchi method with the GRA method has been applied by many researchers, and it has also been used in the field of machining to perform multi-objective optimization [43,44]. Authors such as Mia, Jamil, Pu, Khan and Akhtar et al. optimized the turning constraints using the L27 OA model and ANOVA [45–50]. Singh dealt with turning optimization using the L18OA model and ANOVA [51]. In addition, Vora et al. dealt with optimization when cutting with the L9OA model and ANOVA [52]. Selvan et al. dealt with optimization using the L27OA model and the GRA and ANOVA methods. [53]. Achuthamenon and Sap et al. were concerned with optimization [54,55] when machining composites with the L8OA model and the response surface methodology (RSM) and ANOVA methods.

In our study, we present the evaluation of roundness deviation and the analysis of the effect of roundness deviation on the tool wear parameter and the effect of the quality parameter on the damage parameter of the cutting part of the tool ($VB_N$), in accordance with ISO 3685 [56]. Based on the above literature review and analysis, the motivation of this article is to present the design of a mobile measurement system (MMS) with the new generation of CCHS from the Keyence company (Osaka, Japan), whose use is possible in several production machines [57,58]. Part of the research included the design and manufacture of a holder for CCHS using 3D printing technology. Due to the variety in information on the measurement of surface integrity parameters, we focused on the application of CCHS in the measurement of roundness deviations and linked the results with the $VB_N$ notch wear parameter of the back surface (according to ISO 3685) of the cutting insert when turning C45 steel, as a benchmark of the steel group and at the same time, as one of the most important materials used in the production of shafts for garden equipment [59]. Based on the obtained results, we want to create an information database for the application of CCHS in practice, including the use of the comparison method for machining other materials. Therefore, one of the reasons for choosing the material standard steel C45 is the information obtained from manufacturers who produce products from C45 steel [60,61].

## 2. Materials and Methods

### 2.1. The Experimental System with Confocal Chromatic Sensor (CCHS)

An experimental system (ES) was designed for the study, consisting of a technical system (TS) and a mobile measurement system (MMS), as shown in Figure 1. The TS consisted of a Leadwell T5 CNC machine tool (Leadwell CNC Machines, Taichung City, Taiwan) with a FANUC Oi-MATE-TC control system (FANUC, Yamanashi, Japan). This lathe has a certified maximum runout accuracy of 0.030 mm and a maximum axial runout accuracy of 0.020 mm. A cutting tool with an SSDCN1212K12-M-A holder and SCMTT09T308 TTR cutting insert made of sintered carbide without coating with the following geometry was designed for turning: nose angle $\varepsilon_r = 90°$, main cutting edge setting angle $\kappa_r = 45°$, clearance angle major $\alpha = 7°$ and nose radius $r_\varepsilon = 0.4, 0.8$ and 1.2 mm).

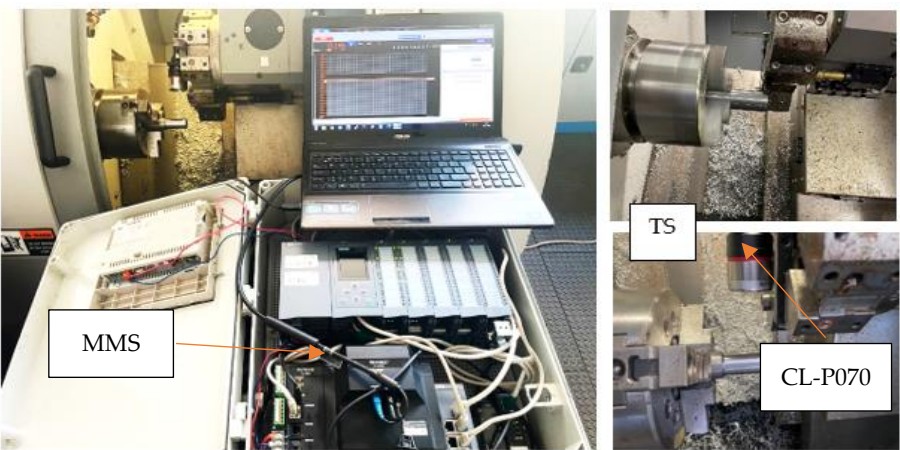

**Figure 1.** Experimental system for turning.

To obtain measurements with a confocal sensor, polychromatic white light from the sensor is projected onto the measured machined surface through a multi-lens system. The lenses are arranged in a confocal arrangement; therefore, when the beam hits the measured object, the radiation is a natural chromatic aberration (deviation of light radiation), divided into monochromatic colors with different wavelengths. The radiation is reflected back to the confocal aperture of the sensor, through which only focused radiation with a specific wavelength passes to the electro-optical sensor. The amount of light that returns to the electro-optical receiver varies significantly, depending on the position of the measured object [62].

MMS consists of a CL-P070 head. The sensor head CL-P070 acquires data from the machined surface. The optical unit CL-P070N is used to process the measured light. The ultra-high-brightness LPD light source produces stable light in all wavelength ranges. The light is received by four high-resolution CMOS crisps, enabling high-precision measurements for all types of materials. The received light is divided by the wavelength per unit distance. The CL-3000 controller transmits the information from the optical unit to external devices, such as a display unit or PC software, using individual ports, such as Profinet, Ethernet and EtherCAT for PLC, or USB for the classic PC connection.

Data transfer from PLC to PC is realized by TCP communication with the TCP server and client. The software on the PC must be started first and must create a TCP server for data acquisition (and wait for data flow). To stream data to the TCP server at a defined interval, the PLC must be started later. The data will be stored in CSV format for later processing.

The C# software starts the TCP server and waits for the connection from the PLC device, retrieves the analyzed data in bytes, and converts the value to the measured value in mm. An example of the main loop of the algorithm with an explanation of the basic principle is shown in Figure 2.

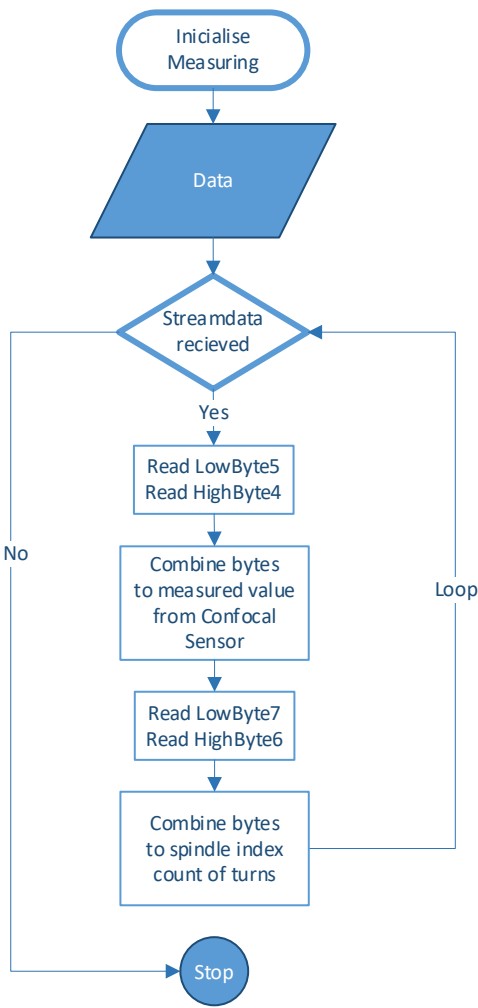

**Figure 2.** The basic principle of the main loop of the algorithm for CCHS.

The measured values are displayed in real-time in the form of a curve, as shown in Figure 3. The communication between the PC and the CL-3000 series Navigator software is carried out via USB.

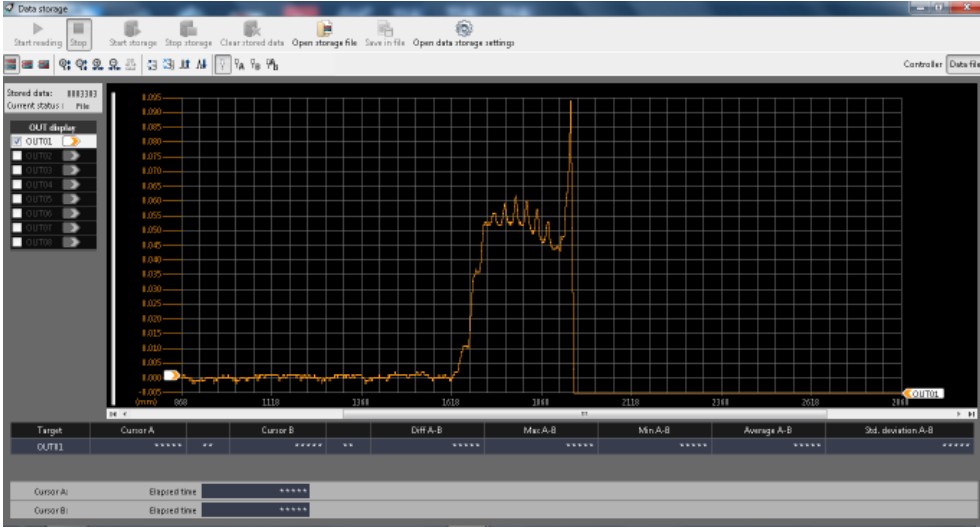

**Figure 3.** Measured data from software in the form of a curve.

The C45 steel with a diameter of 50 mm and a length of 120 mm was chosen for the experiment. The properties and chemical composition of the steel are listed in Table 1.

**Table 1.** Chemical composition of the C45 steel.

| Steel C45 | (%) |
|:---:|:---:|
| C | 0.51 |
| Mn | 0.69 |
| Si | 0.25 |
| Cr | Max. 0.15 |
| Ni | Max. 0.10 |
| P | 0.023 |
| S | 0.017 |

The specimen was mounted at a distance of 55 mm from the face in a 3-jaw chuck. The properties of the material are listed in Table 2 and were verified before the products were manufactured.

**Table 2.** Verified properties of C45 steel products.

| Steel C45 | Values |
|:---:|:---:|
| Density (g/cm$^3$) | 7.85 |
| Hardness HB | Max. 225 |
| Elastic modulus (GPa) | 79 |
| Flexural strength (MPa) | 606 |
| Thermal conductivity (W/mK) | 50 |

The flank wear value ($VB_N$) of the insert was measured and analyzed using a Carl Zeiss Primotech D/A ESD microscope (Zeiss Group, Oberkochen, Germany), as shown in Figure 4.

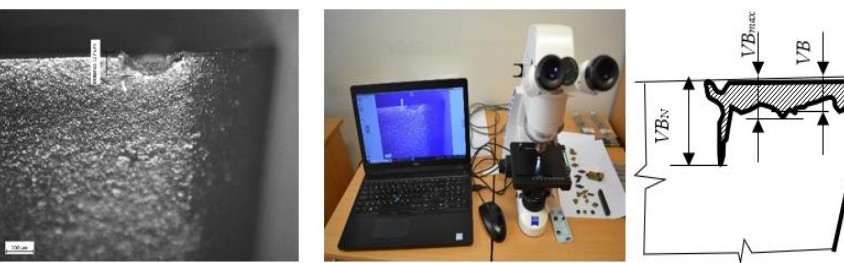

**Figure 4.** Analysis and measurement of the flank wear of the insert ($VB_N$) using a Carl Zeiss Primotech D/A ESD microscope (Zeiss Group, Oberkochen, Germany).

Part of the research included the design and fabrication of a holder for CCHS using 3D printing technology, as shown in Figure 5. The 3D printing technology was developed for the following conditions: printer: Creality Ender 3 (Creality, Shenzhen, China), material: PLA, hot end temperature: 200 °C, infill: 100%, layer height: 0.2 mm, nozzle diameter: 0.4 mm; wall line count: 3 [63].

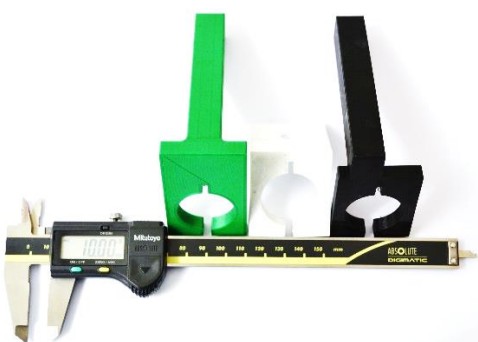

**Figure 5.** The 3D-printed holder for CCHS measuring head.

*2.2. Experimental Matrix and Conditions of Experiments*

The use of orthogonal arrays helps to maximize test coverage by combining inputs and testing the system with comparatively fewer test cases to save time and detect bugs. Based on the analysis of various authors and our research, five input factors at three levels were determined for the L27 orthogonal array.

The conditions of the cutting process are shown in Table 3, with the cutting speed as factor A, feed as factor B, depth of cut as factor C, workpiece attachment length as factor D, and tip radius as factor E. The measurement of roundness deviation was performed using the new-generation CCHS application of CL-P070 [64]. The evaluation of machining conditions for roundness deviation and the effect on tool wear was performed based on the GRA method, in conjunction with the Taguchi method, for the L27 orthogonal array.

The experimental matrix and the results of the initial responses as average values from the measurement are shown in Table 4. The cutting process conditions were recommended based on consultations with product manufacturers and based on experience from previous studies [3]. The parameters for CCHS were set as follows: reference distance of 70 mm, measurement range of ±10.0 mm, spot diameter of 600.0 μm, linearity of ±2.0 μm, resolution of 0.25 μm, optical head weight of 200.0 g, repeatability; number of measurements of 1000 per second.

The roundness deviation $D_R$ of the machined surface was measured and analyzed directly in the working zone of the machine tool with CCHS, i.e., after machining a 50 mm long workpiece without removing it, using a mobile measuring system for CCHS. The roundness deviation was measured along the length of the machined surface (50 mm) in 3 local zones of 5 mm, 25 mm and 45 mm on the cylindrical surface from the end face. The VBN wear was measured after each test. The resulting value of roundness deviation for a given local location was calculated as an average value. The resulting average values of surface roundness deviation ($D_R$) and back surface wear of the cutting insert ($VB_N$) were obtained from 10 repeated measurements, with the minimum and maximum values removed. For each test, a cutting insert with a new cutting edge and without the use of a process medium was used.

**Table 3.** The main machining conditions.

| Factors | Name of Factors | Levels | | |
|---|---|---|---|---|
| | | 1 | 2 | 3 |
| A | Cutting speed (m/min) | 80 | 160 | 240 |
| B | Feed (mm) | 0.1 | 0.2 | 0.3 |
| C | Depth of cut (mm) | 0.05 | 0.1 | 1.0 |
| D | Workpiece attachment length (mm) | 5 | 25 | 45 |
| E | Nose radius (mm) | 0.4 | 0.8 | 1.2 |

**Table 4.** The experimental parameters and results.

| No. | Factors | | | | | Flank Tool Wear (mm) | | Roundness Deviation (mm) | |
|---|---|---|---|---|---|---|---|---|---|
| | A | B | C | D | E | Mean Value (mm) | S/N Ratio | Mean Value (mm) | S/N Ratio |
| 1 | 1 | 1 | 1 | 1 | 1 | 0.121 | 18.3443 | 0.020 | 33.9794 |
| 2 | 1 | 1 | 1 | 2 | 1 | 0.136 | 17.3292 | 0.031 | 30.1728 |
| 3 | 1 | 1 | 1 | 3 | 1 | 0.122 | 18.2728 | 0.032 | 29.8970 |
| 4 | 1 | 2 | 2 | 1 | 2 | 0.142 | 16.9542 | 0.033 | 29.6297 |
| 5 | 1 | 2 | 2 | 2 | 2 | 0.139 | 17.1397 | 0.034 | 29.3704 |
| 6 | 1 | 2 | 2 | 3 | 2 | 0.138 | 17.2024 | 0.032 | 29.8970 |
| 7 | 1 | 3 | 3 | 1 | 3 | 0.156 | 16.1375 | 0.031 | 30.1728 |
| 8 | 1 | 3 | 3 | 2 | 3 | 0.157 | 16.0820 | 0.033 | 29.3704 |
| 9 | 1 | 3 | 3 | 3 | 3 | 0.155 | 16.1934 | 0.033 | 29.3704 |
| 10 | 2 | 1 | 2 | 1 | 3 | 0.145 | 16.7726 | 0.029 | 30.7520 |
| 11 | 2 | 1 | 2 | 2 | 3 | 0.147 | 16.6537 | 0.030 | 30.4576 |
| 12 | 2 | 1 | 2 | 3 | 3 | 0.148 | 16.5948 | 0.031 | 30.1728 |
| 13 | 2 | 2 | 3 | 1 | 1 | 0.151 | 16.4205 | 0.026 | 31.7005 |
| 14 | 2 | 2 | 3 | 2 | 1 | 0.155 | 16.1934 | 0.032 | 29.8970 |
| 15 | 2 | 2 | 3 | 3 | 1 | 0.161 | 15.8635 | 0.032 | 29.8970 |
| 16 | 2 | 3 | 1 | 1 | 2 | 0.144 | 16.8328 | 0.028 | 31.0568 |
| 17 | 2 | 3 | 1 | 2 | 2 | 0.129 | 17.7882 | 0.030 | 30.4576 |
| 18 | 2 | 3 | 1 | 3 | 2 | 0.137 | 17.2656 | 0.030 | 30.4576 |
| 19 | 3 | 1 | 3 | 1 | 2 | 0.162 | 15.8097 | 0.029 | 30.7520 |
| 20 | 3 | 1 | 3 | 2 | 2 | 0.164 | 15.7031 | 0.027 | 31.3727 |
| 21 | 3 | 1 | 3 | 3 | 2 | 0.165 | 15.6503 | 0.031 | 30.1728 |
| 22 | 3 | 2 | 1 | 1 | 3 | 0.157 | 16.0820 | 0.023 | 32.7654 |
| 23 | 3 | 2 | 1 | 2 | 3 | 0.159 | 15.9721 | 0.022 | 33.1515 |
| 24 | 3 | 2 | 1 | 3 | 3 | 0.158 | 16.0269 | 0.021 | 33.5556 |
| 25 | 3 | 3 | 2 | 1 | 1 | 0.171 | 15.3401 | 0.022 | 33.1515 |
| 26 | 3 | 3 | 2 | 2 | 1 | 0.162 | 15.8097 | 0.021 | 33.5556 |
| 27 | 3 | 3 | 2 | 3 | 1 | 0.169 | 15.4423 | 0.021 | 33.5556 |

*2.3. Optimalization of Turning Conditions—Input Factors*

The *S/N* ratio (Equation (1)) was used as a quantitative tool. The higher the ratio, the better. As an output parameter, we proposed a qualitative parameter of the machined surface, namely the roundness deviation. This deviation should be minimal. The following equation applies to the *S/N* ratio:

$$S/N = -10 \log \left[ \frac{1}{n} \left( y_1^2 + y_2^2 + \ldots + y_n^2 \right) \right] \tag{1}$$

where

*S/N* stands for the values of the responses (unit dB);
a $y_1, y_2, \ldots, y_n$ are the observed output values for the test condition repeated n times.

Optimization study and the main effect plots for the *S/N* ratio of roundness deviation and the flank tool wear were performed using MINITAB 20 software. Lower values of $D_R$ surface roundness deviation can improve the operational reliability of the investigated functional surface of the product. At the same time, after the measurement, we received the information that the cutting tool meets the desired technical requirements for the product and does not show undesirable wear. However, a change in the size of the roundness deviation may indicate a change in the cutting tool. Based on this measurement method, we can quickly and accurately detect a localized spot on the functional surface and, at the same time, detect a change in the wear dimension of the cutting tool. These average values of responses depend on specific conditions, so it is necessary to repeat the tests for other materials. We can use the results of the measurements of the machined surfaces on the reference sample of C45 steel to check the surfaces when machining other materials using

the comparative method. These conclusions and results will be reported in the forthcoming publication.

Figures 6 and 7 show the dependence of the mean values of roundness deviation and tool wear. The most important factor that affects the roundness deviation is the cutting speed (as observed in Figure 6a), followed by the depth of cut. Similarly, tool wear was found to be most affected by cutting speed (as shown in Figure 7a), followed by depth of cut. As the depth of cut increases, the roundness deviation increases, causing the surface quality to deteriorate and tool wear to increase. As the cutting speed increases, the roundness deviation decreases, but the risk of damage to the tool increases and wear also increases.

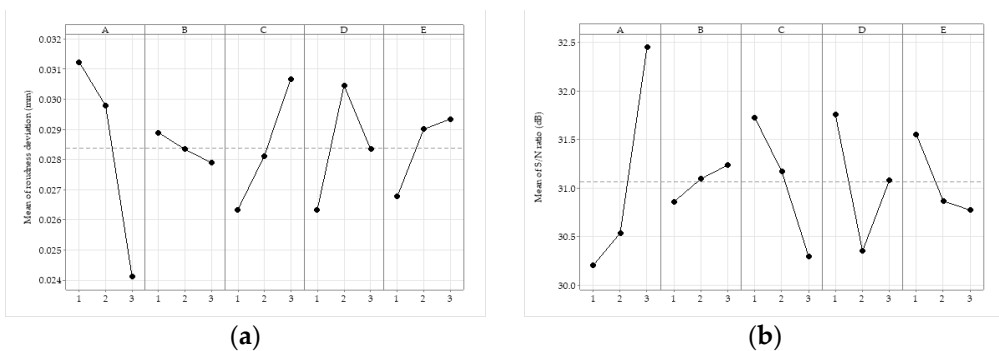

(a)      (b)

**Figure 6.** Main effects plots: (**a**) the effects of input factors on roundness deviation and (**b**) the mean.

*S/N* Ratios Correspond to Roundness Deviation

The ranges between the cutting speed steps 1 and 2 are optimal, which can also be observed in Figures 6b and 7b. Figures 6a and 7a show that the optimal factors and their levels for achieving the responses for roundness deviation are A3B3C1D1E1 and for backside wear are A1B1C1D2E2, which is also shown by the S/N ratio values in Figures 6b and 7b.

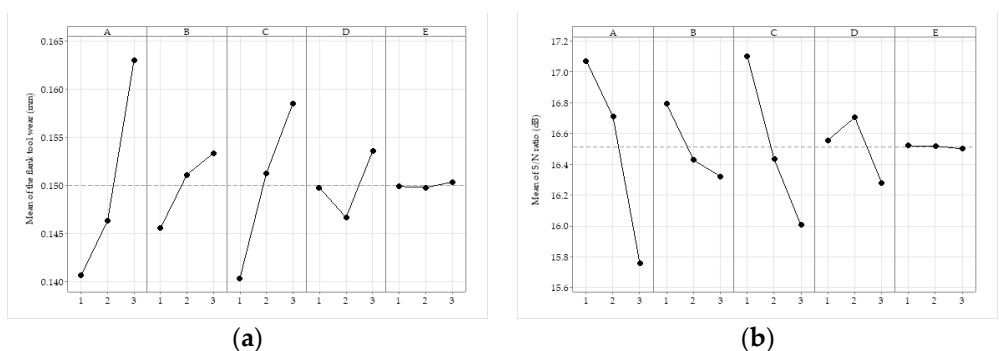

(a)      (b)

**Figure 7.** Main effects plots: (**a**) the effects of input factors on the tool wear of the flank and (**b**) the mean *S/N* ratios corresponding to the tool wear of the flank.

From the results of ANOVA in Table 5, it can be observed that the cutting speed, the depth of cut, and the attachment length of the workpiece are the factors that affect the roundness deviation at the 95% confidence level, as their p-values are less than 0.05, with the cutting speed demonstrating the largest contribution of 45.684%. From Table 6, it can be observed that cutting speed, feed, depth of cut and workpiece attachment length are factors that influence tool wear at the 95% confidence level, with cutting speed demonstrating the largest contribution of 50.584%. The offset and tip radius factors have no significant effect on roundness deviation, and the tip radius factor has no significant effect on the amount of tool wear.

**Table 5.** Results of the ANOVA for roundness deviation.

| Source | Design of Freedom (DF) | Sum of Square (SS) | Means of Square (MS) | F-Value | *p*-Value | Contribution (%) | Remarks |
|--------|------------------------|--------------------|----------------------|---------|-----------|------------------|---------|
| A | 2 | 0.000254 | 0.000127 | 20.09 | 0.002 | 45.684 | Significant |
| B | 2 | 0.000005 | 0.000002 | 0.36 | 0.705 | 0.899 | Insignificant |
| C | 2 | 0.000085 | 0.000043 | 6.75 | 0.007 | 15.288 | Significant |
| D | 2 | 0.000076 | 0.000038 | 6.01 | 0.011 | 13.669 | Significant |
| E | 2 | 0.000035 | 0.000017 | 2.74 | 0.094 | 6.295 | Insignificant |
| Error | 16 | 0.000101 | 0.000006 | | | | |
| Total | 26 | 0.000556 | | | | | |

**Table 6.** Results of the ANOVA for flank tool wear.

| Source | Design of Freedom (DF) | Sum of Square (SS) | Means of Square (MS) | F-Value | *p*-Value | Contribution (%) | Remarks |
|--------|------------------------|--------------------|----------------------|---------|-----------|------------------|---------|
| A | 2 | 0.002426 | 0.001213 | 52.582 | 0.004 | 50.584 | Significant |
| B | 2 | 0.000289 | 0.000144 | 6.264 | 0.010 | 6.026 | Significant |
| C | 2 | 0.001496 | 0.000748 | 32.432 | 0.008 | 31.193 | Significant |
| D | 2 | 0.000214 | 0.000107 | 4.641 | 0.026 | 4.462 | Significant |
| E | 2 | 0.000002 | 0.000001 | 0.032 | 0.967 | 0.042 | Insignificant |
| Error | 16 | 0.000369 | 0.000023 | | | | |
| Total | 26 | 0.004796 | | | | | |

When data from technological processes are processed with different output parameters, the dependence and mutual relationship are complex and very often incomprehensible. We refer to this relationship as gray and it expresses a characteristic of the information that describes this relationship. For the processing of selected results from our research, a procedure was proposed according to the GRA method (Deng [41]). By applying the GRA method, we can express and optimize the incomprehensible relationship of the system for multiple output parameters with the help of a single parameter. This parameter is denoted as GRG (grey relational grade) and is calculated as an average for each output parameter. The result of the optimization of complex multiple factors can be the conversion to a separate (individual) single gray relational grade. Optimizing input machining factors to achieve output parameters using the GRA method requires following the defined procedure in the work of Lin [42]. Gray relational analysis (GRA) was used to optimize the input factors according to [41,42]. After determining the input factors, their levels, and the responses, the design of the factorial matrix (Table 3) and the preprocessing of the data (normalization of the data) follow to reduce their variability. For our responses to be normalized to bring them within an acceptable range, we proposed the method "the smaller the better" and analyzed the data using Equation (2).

$$x_i^*(k) = \frac{maxx_i^0(k) - x_i^0(k)}{maxx_i^0(k) - minx_i^0(k)} \tag{2}$$

where

i = 1, . . . , m and k = 1, . . . , n,
m—the number of experimental data,
n—the number of response characteristics,
$x_i^0(k)$—indicates the original sequence,
$x_i^*(k)$—indicates a sequence after data processing,
$maxx_i^0(k)$—highest value of $x_i^0(k)$,
$minx_i^0(k)$—lowest value of $x_i^0(k)$,

$x_i^0$—is the required value of $x_i^0(k)$.

The next step is to calculate the deviation of the sequence according to (Equation (3)).

$$\Delta_{oi}(k) = |x_o^*(k) - x_i^*(k)| \tag{3}$$

where

$\Delta_{oi}(k)$ is the sequence deviation between the reference sequence $x_0^*(k)$ and the comparison sequence $x_i^*(k)$.

In this study, the reference values of roundness deviation and wear values on the backside of the cutting insert are equal to 1.0. The next step is to calculate the gray relational coefficient (GRC), denoted ξ, according to (Equation (4)). It is an identification coefficient defined in the range $0 \le \xi \le 1$ and depends on the requirements of the system. The determination of this coefficient is important to show the degree of relatedness between the reference sequence $x_0^*(k)$ and the comparison sequences $x_i^*(k)$. Usually, the value of (ξ) is assumed to be 0.5.

$$\xi_i(k) = \frac{\Delta_{\min} + \xi.\Delta_{\max}}{\Delta_{oi}(k) + \xi.\Delta_{\max}} \tag{4}$$

where

i = 1, 2, . . . , m and k = 1, 2, . . . , n;
$\Delta_{\max} = 1.00$ $\Delta_{\min} = 0.00$.

In this step, we calculate the significant indicator gray relational grade (GRG) according to (Equation (5)).

$$\gamma_i = \sum_{k=1}^{n} \omega_1.\xi_i(k) \tag{5}$$

where

$\omega_i$ is the weight of the (i) input variable,
$\gamma_i$ is the required GRG for the (i) experiment,
n is the number of output parameters.

Table 7 shows the resulting values for the GRC and GRG indicators. The multiple optimization problem with multiple responses was transformed into a single optimization indicator (GRG) of the objective function by a combination of the Taguchi method and the GRA method. When the value of the GRG is higher, the corresponding combination of input factors is considered to be near optimal.

**Table 7.** GRC and GRG results for the GRA method.

| Experiment No. | GRC | | GRG | Rank |
|---|---|---|---|---|
| | Roundness Deviation | Flank Tool Wear | | |
| 1 | 1.0000 | 1.0000 | 1.0000 | 1 |
| 2 | 0.3889 | 0.6250 | 0.2493 | 12 |
| 3 | 0.3684 | 0.9615 | 0.3220 | 4 |
| 4 | 0.3500 | 0.5435 | 0.2200 | 18 |
| 5 | 0.3333 | 0.5814 | 0.2243 | 16 |
| 6 | 0.3684 | 0.5952 | 0.2369 | 14 |
| 7 | 0.3889 | 0.4167 | 0.2009 | 22 |
| 8 | 0.3500 | 0.4098 | 0.1889 | 25 |
| 9 | 0.3500 | 0.4237 | 0.1921 | 24 |
| 0 | 0.4375 | 0.5102 | 0.2356 | 15 |
| 11 | 0.4118 | 0.4902 | 0.2241 | 17 |
| 12 | 0.3889 | 0.4808 | 0.2158 | 20 |
| 13 | 0.5385 | 0.4545 | 0.2497 | 11 |
| 14 | 0.3684 | 0.4237 | 0.1971 | 23 |
| 15 | 0.3684 | 0.3846 | 0.1880 | 27 |

**Table 7.** *Cont.*

| Experiment No. | GRC | | GRG | Rank |
| | Roundness Deviation | Flank Tool Wear | | |
|---|---|---|---|---|
| 16 | 0.4667 | 0.5208 | 0.2459 | 13 |
| 17 | 0.4118 | 0.7576 | 0.2862 | 7 |
| 18 | 0.4118 | 0.6098 | 0.2519 | 10 |
| 19 | 0.4375 | 0.3788 | 0.2051 | 21 |
| 20 | 0.5000 | 0.3676 | 0.2193 | 19 |
| 21 | 0.3889 | 0.3623 | 0.1883 | 26 |
| 22 | 0.7000 | 0.4098 | 0.2826 | 9 |
| 23 | 0.7778 | 0.3968 | 0.3004 | 6 |
| 24 | 0.8750 | 0.4032 | 0.3279 | 2 |
| 25 | 0.7778 | 0.3333 | 0.2856 | 8 |
| 26 | 0.8750 | 0.3788 | 0.3222 | 3 |
| 27 | 0.8750 | 0.3425 | 0.3138 | 5 |

To determine the optimal parameters, the GRA method was used in combination with the Taguchi method for the orthogonal array L27. Table 8 lists the optimal factors for achieving higher accuracy of the machined surface (indicated by the roundness deviation) and the effects of each factor on the responses.

**Table 8.** Response table for GRG.

| Process Parameter | GRG—$\gamma_i$ | | |
| | Level 1 | Level 2 | Level 3 |
|---|---|---|---|
| *A* | 0.3149 | 0.2327 | 0.2717 |
| *B* | 0.3177 | 0.2474 | 0.2542 |
| *C* | 0.3629 | 0.2531 | 0.2033 |
| *D* | 0.3251 | 0.2457 | 0.2485 |
| *E* | 0.3475 | 0.2309 | 0.2409 |

Based on the results of the GRG indicator, the order of influence of the levels of individual input factors on the responses is clear and is shown in Table 9. Based on the results (also shown in Table 9), the optimal setting of the input factors is *A*1*B*1*C*1*D*1*E*1.

**Table 9.** The order of influence of the level of input factors *A*, *B*, *C*, *D*, *E* on the responses—$D_R$, $VB_N$.

| Input Factor | Order of Influence of Factor Steps on Roundness Deviation—$D_R$ | Order of Influence of Factor Steps on Wear of Cutting Insert Back Side—$VB_N$ | Order of Influence of Factor Steps on Total GRG |
|---|---|---|---|
| Cutting speed *A* (m/min) | *A*3 > *A*1 > *A*2 | *A*1 > *A*2 > *A*3 | *A*1 > *A*3 > *A*2 |
| Feed *B* (mm) | *B*3 > *B*2 > *B*1 | *B*1 > *B*3 > *B*2 | *B*1 > *B*3 > *B*2 |
| Depth of cut *C* (mm) | *C*1 > *C*2 > *C*3 | *C*1 > *C*2 > *C*3 | *C*1 > *C*2 > *C*3 |
| Workpiece unloading length *D* (mm) | *D*1 > *D*2 > *D*3 | *D*1 > *D*3 > *D*2 | *D*1 > *D*3 > *D*2 |
| Tip radius *E* (mm) | *E*1 > *E*3 > *E*2 | *E*1 > *E*2 > *E*3 | *E*1 > *E*3 > *E*2 |

## 3. Results

The average values from the measurement of the roundness deviation with the CCHS application were compared with the average values of the roundness deviation obtained during the measurement with the device ROUNDTEST RA-120 (Mitutoyo, Kanagawa,

Japan) (Figure 8a,b). The machined surface was measured for the optimum setting of the *A*1*B*1*C*1*D*1*E*1 input factors.

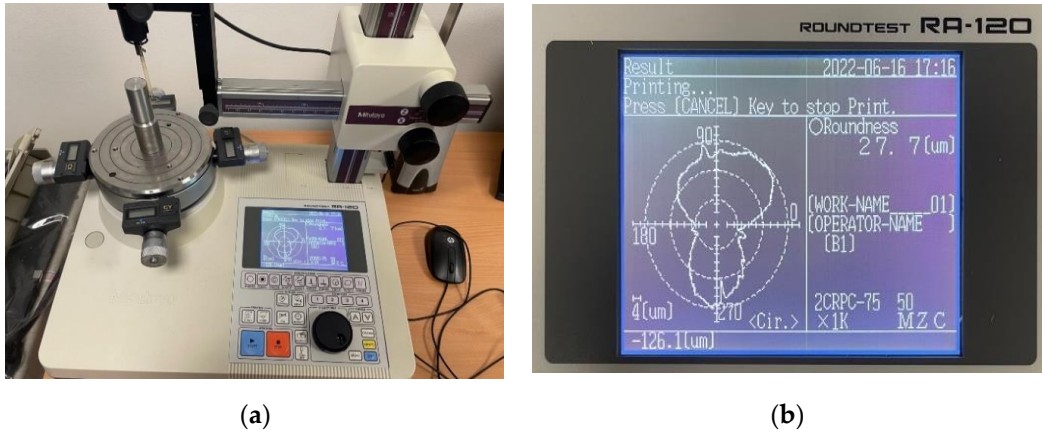

| (**a**) | (**b**) |

**Figure 8.** (**a**) Measurement of roundness deviation using the Roundest RA-120; (**b**) device display with measured curve.

Figure 9a shows the measurement results of the ROUNDTEST RA-120 device (Mitutoyo, Kanagawa, Japan), and Figure 9b shows the measurement results of the CCHS. The measurements have a repeatability value of 5, and the results are average values with subtraction of the minimum and maximum values.

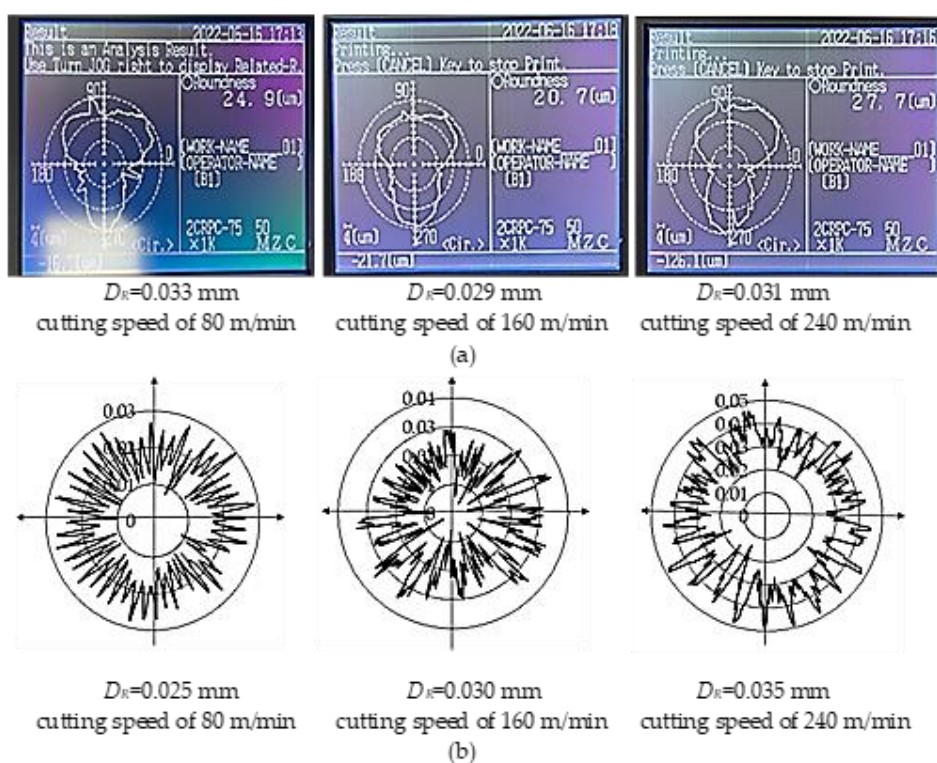

**Figure 9.** (**a**) The measurement results from the ROUNDTEST RA-120 device; (**b**) the measurement results from the CCHS.

From the results of the measurement with the CCHS, which are obtained from ANOVA according to Tables 5 and 6, the cutting speed, the depth of cut, and the attachment length of the workpiece are among the most important factors. Therefore, we focused on these important factors in the comparison of the results, and this comparison is shown in Figure 10a–c.

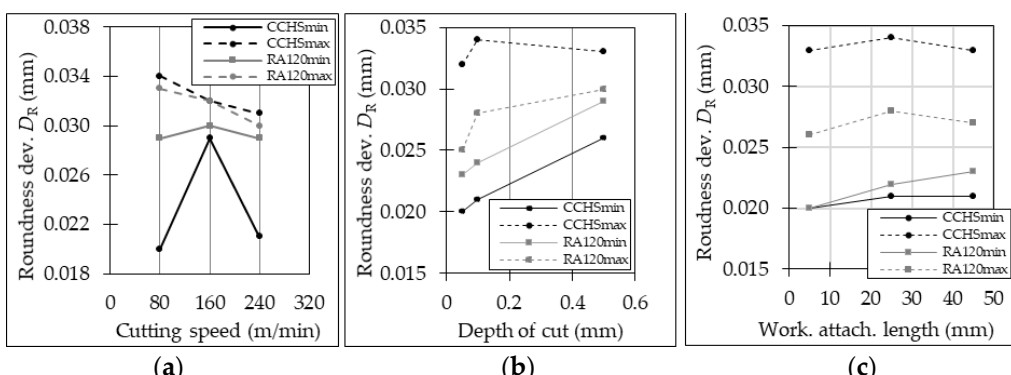

**Figure 10.** Comparison of the resulting values of roundness deviation with the CCHS and on the Roundtest RA-120 device (Mitutoyo, Kanagawa, Japan).

From the results of the comparison of the measurement of the deviation of the roundness of the machined surface, it can be observed that the interval of the values is larger for the measurement with the CCHS than with the RA-120, according to Figure 10a–c. This result was observed for all input factors, such as cutting speed, cutting depth, and workpiece attachment length. The results show that the measurement with CCHS is more accurate and very fast, which was also reported by other authors [10]. The results inform us more accurately about the machined surface, and we can quickly identify local spots where the roundness deviation is outside the specified technical requirements. The quick identification of the accuracy of the local spot is very important, as it reduces the time needed to eliminate possible negative phenomena that can cause a change in the measured deviations. Undesirable phenomena during turning, which can lead to incorrect information about the control of the functional surfaces, mainly include defects on the machined surface (such as particles of stuck chips, cracks, surfaces after friction, traces due to chips coming off, and traces due to a worn tool, etc.), as shown in Figure 11a,b. In this way, we can effectively propose corrections to the technological processes and prevent or avoid the occurrence of defects.

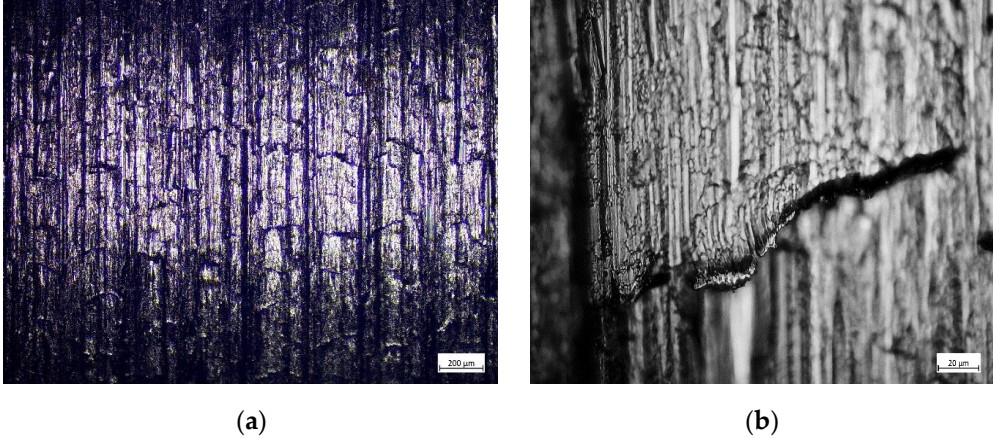

**Figure 11.** Machined surface after turning and negative phenomena: (**a**) turning surface, (**b**) traces due to chips coming off, (80 m/min, feed: 0.1 mm, depth of cut: 0.05 mm, workpiece attachment length: 5 mm, and tip radius: 0.4 mm).

## 4. Discussion

In this paper, the suitability of using a confocal chromatic sensor (CCHS) for evaluating the quality of a machined surface, in particular the accuracy of the roundness deviation of the cylindrical surface, was demonstrated. The results obtained were evaluated using the gray relational analysis method (GRA) in conjunction with the Taguchi method, and the significance of the factors was demonstrated using the ANOVA method. Part of the study

also included a comparison of the data on the circular shape deviation measured with the device Roundtest RA 120 (Mitutoyo, Kanagawa, Japan) and with the CCHS. The results with the reference material C45 steel show that the factors of cutting speed, depth of cut and workpiece attachment length influence the roundness deviation and that the factors of cutting speed, feed, depth of cut, and workpiece attachment length, in conjunction with each other, influence the wear on the back of the tool. The roundness deviation is smaller with increasing cutting speed and feed and decreasing depth of cut. Larger values of roundness deviation were found at lower cutting speeds and feeds and lower values of depth of cut. The VBN parameter is mainly negatively affected by higher cutting speeds, higher feeds, and greater depths of cut. In connection with the research on negative phenomena, such as the wear of the cutting tool and its effect on the circularity deviation, the GRA Taguchi method was applied, and the result of the optimization (according to Tables 7 and 8) for the circularity deviation and tool wear was the determination of the optimal input factors for C45 steel. The results show the validity of conducting tests of various combinations of input factors, as well as the proposal of a combination of various output parameters for the determination of theoretical models for practical needs.

## 5. Conclusions

The following conclusions can be drawn from the research analysis. The Taguchi method for the L27 orthogonal array was used to optimize the following input factors: cutting speed, denoted as factor *A*, feed as factor *B*, depth of cut as factor *C*, workpiece attachment length as factor *D*, and tip radius as factor *E*. It was also used to investigate the effect on the output factors (roundness deviation and cutting tool wear on the back surface) separately. Figures 6 and 7 present the result of the input parameters for the individual output factors. The ANOVA method was used to analyze the significance of the input parameters, and the results are shown in Table 5 for roundness deviation and in Table 6 for the cutting tool wear on the back surface. The result of the optimization of the Taguchi method for the L27 orthogonal array was the design of the input parameters A3B3C1D1E1 for the roundness deviation and A1B1C1D2E2 for the cutting tool wear on the back surface. The effects of process input factors, such as cutting speed, feed, depth of cut, workpiece attachment length, and tip radius, on the responses for roundness deviation $D_R$ and cutting tool wear $VB_N$, measured after turning the machined surface of C45 steel with CCHS, were investigated using GRA, in conjunction with the Taguchi method, and the following conclusions were drawn. From the main effect diagrams and *S/N* ratios, it can be observed that the most influential factor for $VB_N$ is cutting speed, followed by the depth of cut, feed, and workpiece attachment length. The most influential factor for DR is the cutting speed, followed by the depth of cut and the workpiece attachment length. A reduction in $D_R$ and $VB_N$ can be achieved with a shallower depth of cut (i.e., a smaller allowance or cross-section of the removed layer) and a higher cutting speed. From the analysis of the ANOVA, it is clear that the influence of process factors on $D_R$ is different. The influence of cutting speed and depth of cut is 50.584% and 31.193% for $VB_N$, respectively. The influence of feed and attachment length of the workpiece is less than 8%. The percentage of influence of the cutting speed, depth of cut and attachment length for $D_R$ is 45.684%, 15.288% and 13.669%, respectively. The influence of displacement and tip radius is less than 8%. For $VB_N$ wear, the influence of tip radius is small and for roundness deviation, the influence of feed is small. The optimum condition for achieving lower $VB_N$ and $D_R$ wear is based on the results of the GRA in conjunction with the Taguchi method and is *A1B1C1D1E1*, i.e., a cutting speed of 80 m/min, feed of 0.1 mm, depth of cut of 0.05 mm, workpiece attachment length of 5 mm and tip radius of 0.4 mm. The difference in the optimization process using the GRA method is that we examined two output factors together (between which the relationship is incomprehensible and unclear) and the result of the optimization is the degree of GRG, which expresses this relationship. The result of optimization using the GRA method is the design of the input parameters as *A1B1C1D1E1*. A combination of factors determines the damage mode on the back side of the cutting insert and then

influences the roundness deviation and the turning process. Provided that the turning conditions (setting of optimum conditions), the conditions of the technological system (i.e., the prescribed tolerances of deviations due to vibrations), and the measurement conditions (i.e., the reference and optimum conditions for measurement) are maintained, it is possible to use the CCHS for measuring qualitative parameters of the machined surface. Based on the obtained results, our intention is to use these data to create a model for the application of laser sensors in practice, and in further research, to use the comparison method for measuring surface integrity parameters when machining other materials. The use of laser sensors for product measurement directly in the work zone after machining is of great importance and brings many advantages, such as product measurement directly on the machine, high measurement speed and comprehensive information about the machined surface. Examining the obtained results on the machined surface profile (defined by different curves) and determining the conditions for achieving accuracy with determined repeatability of the data are other areas of current research.

**Author Contributions:** Conceptualization, J.J. and M.M.-P.; methodology, J.J.; software, M.M.-P.; validation, J.J. and J.H.; formal analysis, M.M.-P.; investigation, J.J.; resources, P.M.; data curation, M.M.-P.; writing—original draft preparation, J.J.; writing—review and editing, J.H.; visualization, M.M.-P.; supervision, P.M. and J.J.; project administration, J.H.; funding acquisition, J.H. All authors have read and agreed to the published version of the manuscript.

**Funding:** This research was funded by the projects KEGA 038TUKE-4/2022 granted by the Ministry of Education, Science, Research and Sport of the Slovak Republic.

**Data Availability Statement:** Not applicable.

**Acknowledgments:** As the authors of the article, we would like to thank the research team of the progressive production technologies for the support of research works by the grant agency APVV-19-0590 and also the projects VEGA 1/0268/22 and KEGA 038TUKE-4/2022, supported by the Ministry of Education, Science, Research and Sport of the Slovak Republic.

**Conflicts of Interest:** The authors declare no conflict of interest.

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
