# Peer review of "Turned Surface Monitoring Using a Confocal Sensor and the Tool Wear Process Optimization"

_processes, doi:10.3390/pr10122599_

Round 1
Reviewer 1 Report
- The English can be improved. So many direct speeches seem monotonous
- The justification for the selection of method is missing i.e why L27 OA is taken, and why GRA was chosen. Those are not clearly explained
- How the CCHS work and help the engineers in understanding the process behavior or output should be explained
- Fig 6 is incorrect and SN ratio is missing
- please explain the difference between the optimization carried out using the simile Taguchi method and GRA Taguchi method. Does it have any different conclusion?
- The end section of experimental results and conclusions are not very clear. Please highlight the method used would provides improved results by the use of instruments or the experimental technique. Validation with some existing study would improve its acceptability.
-Calibration of the CCHS system is important and what is the accuracy and repeatability need to be explained
- For the selection of SN ratio why smaller the better is taken or larger the better is taken
Author Response
Thank you for your review. The answered questions are in the appendix.

Reviewer 2 Report
This paper investigates the evaluation of roundness deviation and tool wear by using confocal chromatic sensor (CCHS), the optimized processing parameters are also given. The work is interesting and meaningful, and it can be published after revision. Some suggestions are given as follows:
1. Fig. 2 is not appropriate for a research article, and it can be replaced by algorithm flow.
2. The font size in Figs. 6 and 7 are too small, in addition, the axial label should be added.
3. The quality of Fig. 9 should be improved.
4. The full name of ANOVA should be given in abstract.
5. The content of the discussion part is not enough, and the authors can discuss the obtained effects of processing parameters on roundness deviation and tool wear in theory, which is important for research.
6. As the author said in the first paragraph, machining is almost irreplaceable, and some references about machining can be added:
1) Introduction to high-speed machining (HSM). in High Speed Machining, New York, NY, USA: Academic, 2020: 1–25.
2) Developing a ball screw drive system of high-speed machine tool considering dynamics. IEEE Transactions on Industrial Electronics, 2022, 69(5): 4966-4976.
Author Response

(The authors gave the same response as above.)

Reviewer 3 Report
Manuscript ID processes – 2044490
Title: «Turned Surface Monitoring Using a Confocal Sensor and the Tool wear Process Optimization».
Overall, the topic of this paper is relevant, and the manuscript was good organized and written. Undoubtedly, the presented manuscript is relevant from a scientific and practical point of view. This study opens up defined prospects in this field of knowledge. Manuscript entitled "Turned Surface Monitoring Using a Confocal Sensor and the Tool wear Process Optimization" of interest for a highly ranked journal like "Processes".
This manuscript includes the next harmonious structure:
1. Introduction (p. 1 – 3);
2. Materials and Methods (p. 3 – 12);
3. Results (p. 12 – 14);
4. Discussion (p. 14);
5. Conclusions (p. 14 – 15).
Methodological inaccuracies are not detected. The figures and tables are appropriate, they reflect the complex results of this study. So, they easy to interpret and understand by readers. References list is adequate and includes 61 titles.
The value of this work is significant, but I hope that next suggestions can help to improve the manuscript.
1) Please make clearer the aim of the study. In my opinion, general aim of the study is not formulated.
2) Indicate the directions of further research or improvements based on Taguchi method.
Author Response

(The authors gave the same response as above.)

Round 2
Reviewer 1 Report
The paper is improved a lot. Still this point is not discussed.
-In some cases the figures are not readable.
- Please explain the difference between the optimization carried out using the simple or only Taguchi method (multi-response with multiple input parameters) and GRA Taguchi method. Does it have any different conclusions?
Author Response
Thank you for your review.

Reviewer 2 Report
No comments.
Author Response
Thank you for your review.
